# A toolkit for studying cell surface shedding of diverse transmembrane receptors

Amanda N Hayward, Eric J Aird, Wendy R Gordon*

Department of Biochemistry, Molecular Biology, and Biophysics, University of Minnesota, Minneapolis, United States

**Abstract** Proteolysis of transmembrane receptors is a critical cellular communication mechanism dysregulated in disease, yet decoding proteolytic regulation mechanisms of hundreds of shed receptors is hindered by difficulties controlling stimuli and unknown fates of cleavage products. Notch proteolytic regulation is a notable exception, where intercellular forces drive exposure of a cryptic protease site within a juxtamembrane proteolytic switch domain to activate transcriptional programs. We created a Synthetic Notch Assay for Proteolytic Switches (SNAPS) that exploits the modularity and unequivocal input/response of Notch proteolysis to screen surface receptors for other putative proteolytic switches. We identify several new proteolytic switches among receptors with structural homology to Notch. We demonstrate SNAPS can detect shedding in chimeras of diverse cell surface receptors, leading to new, testable hypotheses. Finally, we establish the assay can be used to measure modulation of proteolysis by potential therapeutics and offer new mechanistic insights into how DECMA-1 disrupts cell adhesion.
DOI: https://doi.org/10.7554/eLife.46983.001

## Introduction

Proteolysis of cell surface transmembrane proteins is a tightly regulated cellular mechanism that controls communication of cells with their extracellular environment. Diverse adhesion receptors, such as cadherins, as well as receptors that respond to soluble factors, such as receptor tyrosine kinases (RTKs), have been shown to be cleaved at sites close to the extracellular side of the membrane by metalloproteinases such as 'A Disintegrin And Metalloproteinases' (ADAMs) and matrix metalloproteinases (MMPs) (*Kessenbrock et al., 2010*; *McCawley and Matrisian, 2001*; *Miller et al., 2017*; *Seals and Courtneidge, 2003*; *White, 2003*), resulting in ectodomain shedding. In many of these receptors, ectodomain shedding is a prerequisite for further cleavage inside the membrane by the γ-secretase/presenilin protease complex in a process known as Regulated Intramembrane Proteolysis (RIP) (*Brown et al., 2000*; *Selkoe and Wolfe, 2007*).

Proteolysis not only provides a mechanism to break cell-cell and cell-ECM contacts to modulate processes such as cell migration, but may also result in biologically-active fragments outside and inside of the cell, such as the intracellular fragment of Notch, which translocates to the nucleus and acts as a transcriptional co-activator (*Bray, 2006*; *Struhl and Adachi, 1998*; *Struhl and Greenwald, 1999*). Dysregulated proteolysis contributes to disease pathogenesis, for example, by causing accumulation of pathogenic fragments such as the amyloid beta peptide implicated in Alzheimer's disease (*Goate et al., 1991*; *Scheuner et al., 1996*), or removing epitopes required for normal cell communication (*Boutet et al., 2009*; *Groh et al., 2002*; *Kaiser et al., 2007*; *Waldhauer et al., 2008*). For instance, cancer cells evade the immune response by shedding MICA receptors (*Boutet et al., 2009*; *Groh et al., 2002*; *Kaiser et al., 2007*; *Waldhauer et al., 2008*), which are normally deployed to the cell surface in response to cellular damage.

*For correspondence:
wrgordon@umn.edu

Competing interests: The authors declare that no competing interests exist.

Modulation of proteolysis is a heavily pursued therapeutic avenue, aiming to either inhibit proteases or prevent access to protease sites in a specific receptor. Many protease inhibitors have been developed but have failed clinically due to significant off-target effects (*Dufour and Overall, 2013*; *Turk, 2006*; *Vandenbroucke and Libert, 2014*). Conversely, relatively few examples of modulating access to protease sites in specific receptors have been reported, despite the clinical success of the monoclonal antibody trastuzumab (Herceptin) that was found to act, in part, by blocking proteolysis of the receptor tyrosine kinase HER2 (*Molina et al., 2001*). Similarly, successful development of modulatory antibodies targeting proteolysis of Notch (*Agnusdei et al., 2014*; *Aste-Amézaga et al., 2010*; *Falk et al., 2012*; *Li et al., 2008*; *Qiu et al., 2013*; *Tiyanont et al., 2013*; *Wu et al., 2010*) and MICA (*Ferrari de Andrade et al., 2018*) receptors have recently been reported. However, although 8% of the annotated human transmembrane proteins are predicted to be shed from the surface (*Tien et al., 2017*), mechanisms of proteolytic regulation that inform development of specific modulators have remained elusive.

A relatively unique proteolytic regulation mechanism has recently come to light in which a stimulus alters protein conformation to induce exposure of a cryptic protease site. For example, the secreted von Willebrand factor (VWF) is cleaved in its A2 domain in response to shear stress in the bloodstream, which regulates blood clotting (*Dong et al., 2002*). Transmembrane Notch receptors also control exposure of a cryptic protease recognition site via the conformation of a juxtamembrane domain called the Negative Regulatory Region (NRR) (*Gordon et al., 2015*; *Gordon et al., 2009*; *Gordon et al., 2007*; *Sanchez-Irizarry et al., 2004*; *Xu et al., 2015*) to trigger Notch signaling (*Bray, 2006*; *Kopan and Ilagan, 2009*) in response to ligand binding and subsequent endocytic forces (*Gordon et al., 2015*; *Langridge and Struhl, 2017*; *Parks et al., 2000*). Crystal structures of the NRR (*Gordon et al., 2009*; *Gordon et al., 2007*; *Xu et al., 2015*) reveal that the ADAM10/17 protease site is housed in a Sea urchin Enterokinase Agrin-like (SEA-like), with high structural homology to canonical SEA domains of mucins (*Macao et al., 2006*; *Maeda et al., 2004*) but lacking the characteristic autoproteolytic site. Canonical SEA domains of mucins contain a highly conserved αβ-sandwich ferredoxin fold despite low sequence conservation (*Macao et al., 2006*). Notch's SEA-like domain was identified based on structural homology (*Gordon et al., 2007*) and a recent computational study identified several previously unidentified juxtramembrane domains predicted to exhibit an SEA-like fold (*Pei and Grishin, 2017*).

The NRR normally exists in a proteolytic cleavage-resistant state in which the protease site is buried by interdomain interactions between the SEA-like and its neighboring domain but can be switched to a protease-sensitive state when it undergoes a conformational change upon binding a ligand on a neighboring cell and subsequent trans-endocytosis (*Gordon et al., 2015*; *Parks et al., 2000*; *Sanchez-Irizarry et al., 2004*) or if it harbors disease-related mutations that destabilize the domain (*Gordon et al., 2009*; *Malecki et al., 2006*; *Weng et al., 2004*).

Notch's proteolytic switch has been exploited to develop conformation-specific modulatory antibodies (*Agnusdei et al., 2014*; *Aste-Amézaga et al., 2010*; *Falk et al., 2012*; *Li et al., 2008*; *Qiu et al., 2013*; *Tiyanont et al., 2013*; *Wu et al., 2010*) and harnessed for synthetic biology applications (*Morsut et al., 2016*) to turn on transcription in response to any desired cell-cell contact. For example, Notch was engineered to respond to novel inputs and create custom responses (*Roybal et al., 2016*). This SynNotch system has been applied to CAR T-cell therapy to require dual antigen recognition for T-cell activation, increasing specificity. Thus, identification of additional proteolytic switches is of great interest. However, despite the knowledge that several cell-surface receptors harbor extracellular juxtamembrane domains with structural homology to Notch's proteolytic switch (*Pei and Grishin, 2017*) and that more than 100 receptors undergo a Notch-like proteolytic cascade (*Brown et al., 2000*; *Selkoe and Wolfe, 2007*), other membrane resident proteolytic switches have not been identified, in large part due to difficulties in studying proteolysis in most receptors. For example, controlling the stimulus for receptors involved in homotypic interactions is difficult and the signaling pathways modulated by cleaved intracellular fragments may not be known.

A recent study showing that the known VWF proteolytic switch domain could functionally substitute for the Notch NRR to facilitate Notch signaling in certain contexts in *Drosophila* (*Langridge and Struhl, 2017*) inspired us to ask if we could exploit Notch signaling to discover new proteolytic switches. We created a Synthetic Notch Assay for Proteolytic Switches (SNAPS) that harnesses the modularity and precise control of Notch signaling (*Gordon et al., 2015*; *Malecki et al., 2006*;

*Roybal et al., 2016*) to screen protease site-containing juxtamembrane domains of diverse cell-surface receptors for their ability to functionally substitute for Notch's proteolytic switch and induce transcription in response to cell-cell contact. SNAPS uses the native Notch ligand-binding interaction with DLL4 as the input and the Gal4 transcriptional response as the output (*Figure 1A*). Here, we find that proteolysis regions of several receptors with structural homology to Notch can substitute for the Notch 'proteolytic switch' and facilitate signaling in response to cell contact. Moreover, the assay can be used to detect shedding of diverse receptors such as RTKs and cadherins. Finally, we demonstrate that the assay can be used to screen modulators of proteolysis.

## Results

### SEA-like domains cooperate with adjacent domains to behave as proteolytic switches

To determine if receptors bearing juxtamembrane domains predicted to be structurally homologous to Notch could also function as proteolytic switches, we created chimeric receptors in which we replaced the Notch NRR proteolytic switch domain with recently identified/predicted SEA and SEA-like domains from other cell surface receptors and included relevant tandem N-terminal domains (*Figure 1B*, *Supplementary File 1*). We hypothesized that other putative proteolytic switches could functionally substitute for the Notch NRR and initiate a transcriptional response in response to contact with a cell expressing DLL4. We also made a negative control chimera where the NRR was replaced by the fluorescent protein mTFP. In the SNAPs assay, chimeric constructs together with Gal4-responsive and control luciferase reporter constructs were transfected into U2OS cells, co-cultured with cells stably expressing Notch ligands, and luciferase activity measured in a high-throughput format.

Surprisingly, we found that the ECM receptor dystroglycan and two protocadherins involved in intercellular adhesion, Protocadherin-15 (PCDH15) and Cadherin-related protein 2 (CDHR2), could functionally substitute for Notch's NRR (*Figure 1C*). These chimeric receptors signaled robustly only in the presence of cells expressing DLL4, and the signal was abrogated by both a global metalloproteinase inhibitor (BB-94) and an inhibitor of the subsequent intramembrane γ-secretase cleavage event (γ-secretase inhibitor; GSI). The putative cell adhesion molecules Trop2 and Cadherin-23 (CADH23) displayed a more moderate signaling activity in response to DLL4. Interestingly, all of these receptors contain an SEA or an SEA-like domain in tandem with an N-terminal domain. On the other hand, SEA/SEA-like domains without a structured neighboring domain, such as Mucin-1 (MUC1) and receptor tyrosine phosphatase-related islet antigen 2 (IA-2), exhibited a high level of proteolysis even in the absence of DLL4, suggesting they contain a constitutively exposed protease site in the context of this assay.

A few chimeras showed very little signal in the assay (*Figure 1—figure supplement 1*), suggesting a lack of proteolysis or lack of cell-surface expression. To further probe the receptors exhibiting low levels of activation in the signaling assay, we performed a cell-surface ELISA assay. Briefly, Flag-tagged Notch chimera constructs were transfected into U2OS cells, fixed, stained, and cell-surface levels quantified by measuring HRP activity. Most of the chimeras lacking signaling activity also expressed at lower levels than Notch, suggesting defects in expression or trafficking due to incorrect choice of domain termini. Our negative control mTFP chimera and MUC13 expressed substantially better than Notch (*Figure 1D*), suggesting lack of response in the signaling assay is due to an absence of proteolysis in the assay. In contrast, the ELISA showed that IA-2 expressed at much lower levels than Notch yet exhibited robust constitutive signaling activity. We reasoned that high rates of shedding could result in apparently low cell-surface levels in the ELISA assay, so we repeated the ELISA assay with the addition of the metalloproteinase inhibitor BB-94. Indeed, IA-2 surface expression was substantially increased in the presence of BB-94 (*Figure 1—figure supplement 2*), while surface levels of other receptors that exhibited constitutive signaling activity were not drastically affected by inhibitor treatment. This suggests that IA-2 undergoes much higher rates of proteolysis than the other proteins studied. Since we observed variable cell-surface levels of the receptors in the ELISA assay, we also performed titrations of the chimeric receptors in the SNAPS signaling assay to ensure that high surface level expression was not masking proteolytic switch-like behavior (*Figure 1—figure supplement 3*). We generally recommend titration of chimera concentration to avoid

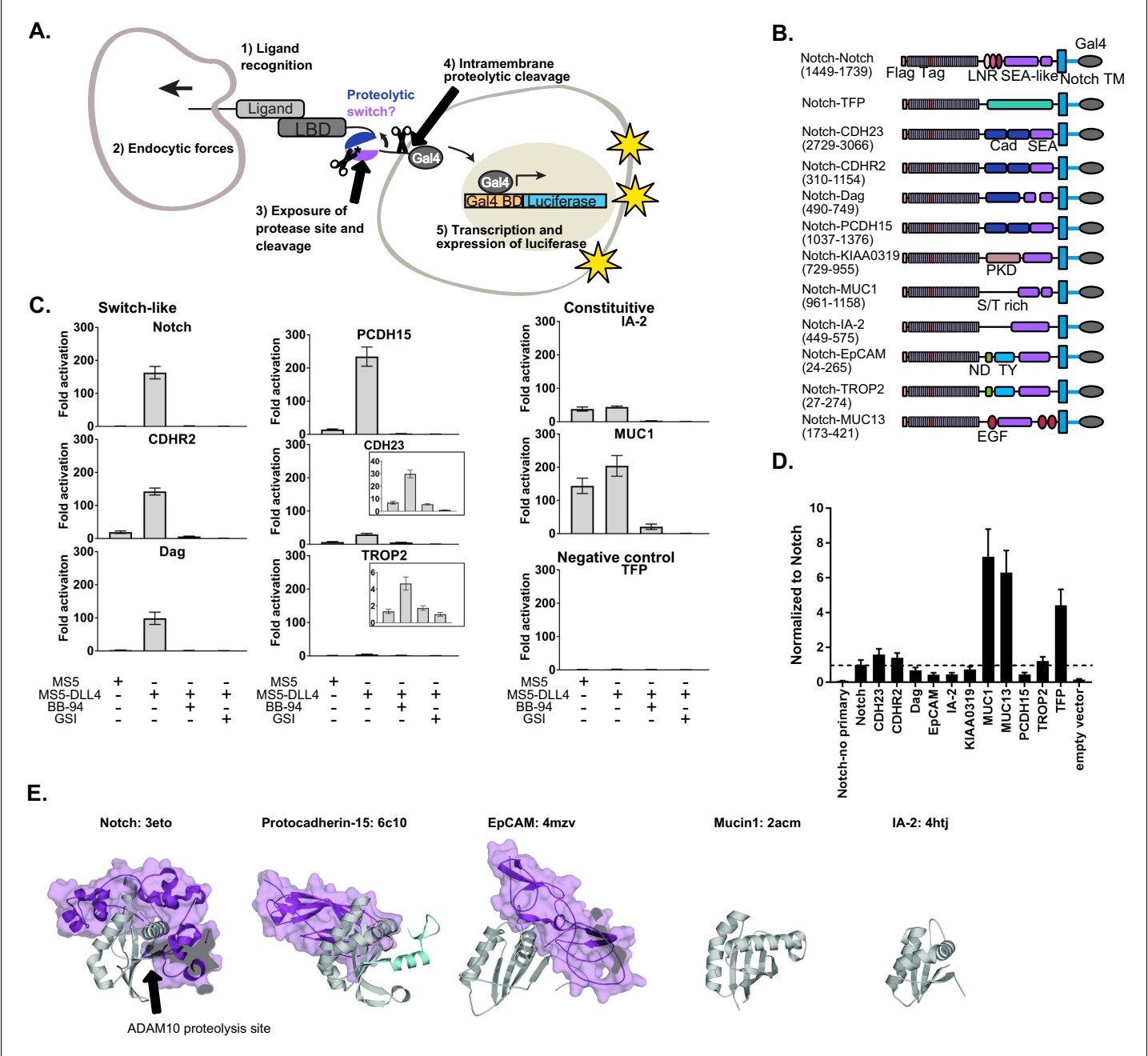

**Figure 1.** SEA-like domains cooperate with adjacent domains to behave as proteolytic switches. (A) Schematic of Synthetic Notch Assay for Proteolytic Switches (SNAPS). Cells co-expressing Flag-Notch-X-Gal4 chimeras, where X is a putative proteolysis region of another receptor, and luciferase reporter constructs are co-cultured with DLL4 ligand-expressing cells to induce Notch activation and expression of luciferase. (B) Schematic of chimeric constructs utilized in the signaling assay. Protein domains are color coded and labeled below. Amino acid ranges used for each construct are in parentheses under the names. Note that Notch's SEA-like domain is also referred to as the Heterodimerization Domain (HD) in the literature. Abbreviations used: Cad: cadherin. EGF: Epidermal growth factor. LBD: Ligand binding domain. LNR: Lin-12 Notch-like repeats. ND: N-terminal domain. PKD: polycystic kidney disease domain. S/T rich: serine-threonine rich. TFP: Teal fluorescent protein. TM: transmembrane domain. TY: thyroglobulin type-1A domain. (C) Luciferase reporter gene activity profile of Notch and Notch chimera constructs (1 ng transfected in 96wp) co-cultured with MS5 cells or MS5 cells stably expressing DLL4. BB-94 = Batimastat (pan-metalloproteinase inhibitor) GSI = Compound E (γ-secretase inhibitor). Data shown are triplicate measurements from a representative experiment. Error bars represent the SEM of triplicate measurements. (D) Cell surface ELISA of Notch and Notch chimera constructs. Anti-Flag primary and goat anti-mouse HRP secondary antibodies were used to detect cell surface expression levels of each chimera. The horizontal dotted line corresponds to Notch expression levels. Error bars represent the SEM of triplicate measurements. (E) Structures and PDB IDs of SEA-like domains (gray) with applicable adjacent domains (purple). The Notch adjacent domain is

*Figure 1 continued on next page*

*Figure 1 continued*

comprised of three cysteine-rich, calcium binding Lin12 Notch repeats. Protocad15 (*De-la-Torre et al., 2019*; *Dionne et al., 2018*; *Ge et al., 2018*) has an Ig-like adjacent domain and EpCAM (*Pavšič et al., 2014*) has a cysteine-rich thyroglobulin adjacent domain. The buried surface area of the adjacent domains are 3800, 1300, and 2800 square Angstroms for Notch, Protocad15, and EpCAM, respectively. SEA-like domains were structurally aligned to the Notch SEA-like domain.

DOI: https://doi.org/10.7554/eLife.46983.002

The following figure supplements are available for figure 1:

**Figure supplement 1.** SEA domain chimeras without signaling activity.

DOI: https://doi.org/10.7554/eLife.46983.003

**Figure supplement 2.** ELISA in the presence of BB-94.

DOI: https://doi.org/10.7554/eLife.46983.004

**Figure supplement 3.** Titration of DNA used in co-culture assay.

DOI: https://doi.org/10.7554/eLife.46983.005

**Figure supplement 4.** Plated ligand assay.

DOI: https://doi.org/10.7554/eLife.46983.006

**Figure supplement 5.** Shedding of diverse receptors detected by SNAPS.

DOI: https://doi.org/10.7554/eLife.46983.007

effects due to abnormally high cell surface expression that can mask ligand-induced effects. Most receptors showed decreasing signaling activity with decreasing concentration of receptor, as expected. Interestingly, IA-2 signal increased as receptor concentration decreased, perhaps related to its high expression levels and turnover rates. SNAPS also works in the context of plated recombinant DLL4 ectodomain (*Figure 1—figure supplement 4*), a common mode of performing the Notch signaling assay, which may be more convenient in some cases.

Comparing solved structures of several SEA/SEA-like domain containing proteins reveals additional insights (*Figure 1E*). SEA/SEA-like domains are colored gray with adjacent N-terminal domains in purple. In contrast to Notch, Protocadherin-15, EpCAM, MUC1, and IA-2 do not have structured domains N-terminal to their SEA/SEA-like domain. This likely leads to enhanced conformational dynamics, resulting in an increase in protease site exposure and signaling. Although Notch and Protocadherin-15 exhibit similar conformational switch behavior in the signaling assay, Protocadherin-15's N-terminal cadherin-like domain binds on the opposite face of the SEA-like domain than Notch's neighboring Lin12 Notch Repeat domain, suggesting potentially different conformational switching propensities.

## SNAPS to probe MMP proteolysis of dystroglycan

We next aimed to investigate whether this assay can be used effectively to test hypotheses about regulation and potential modulation of proteolysis by the conformation of SEA-like domains. We chose to further investigate the extracellular matrix receptor dystroglycan, which provides a critical mechanical link between the ECM and the actin cytoskeleton to provide stability to muscle cells and maintain the blood–brain barrier (*Agrawal et al., 2006*; *Barresi and Campbell, 2006*), as a model because the conformational regulation of proteolysis of the native dystroglycan protein in vitro has been recently explored in parallel studies (*Hayward and Gordon, 2018*). This study showed that dystroglycan containing an intact proteolytic switch domain exhibited low levels of proteolysis but that destabilized proteolytic switch domains via mutation or truncation displayed enhanced proteolysis. Moreover, degrees of proteolysis observed correlated with alterations in cell migration phenotypes (*Hayward and Gordon, 2018*), an important finding given that dystroglycan proteolysis by MMPs is enhanced in many pathogenic states (*Agrawal et al., 2006*; *Matsumura et al., 2005*; *Singh et al., 2004*).

Thus, we used SNAPS to determine whether the Notch-dystroglycan chimera containing the entire proteolytic switch domain is protected from proteolysis when exogenous MMPs are added. For comparison, we also measured proteolysis in deletion chimera (ΔCadΔSEA) expected to have constitutively exposed protease sites (*Figure 2A*). Indeed, while the chimera containing constitutively exposed protease sites displayed robust 30-fold activation in the SNAPS assay upon addition of MMPs, the chimera containing an intact proteolysis domain was substantially protected from MMP cleavage, showing only a two- to three-fold increase in signal when MMPs were added

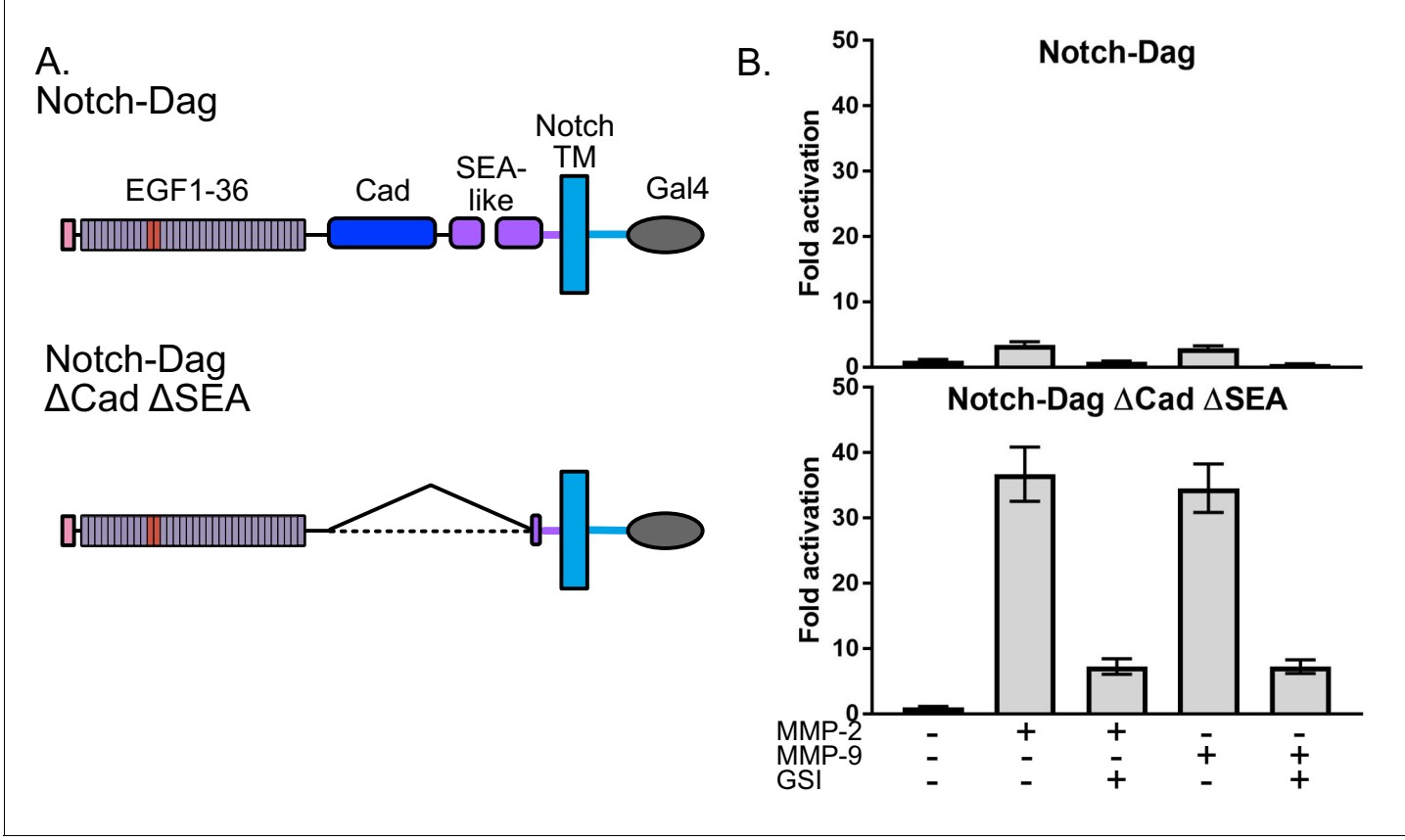

**Figure 2.** Dystroglycan containing intact proteolytic switch domain is protected from MMP cleavage. (**A**) Chimera constructs used to test MMP sensitivity of Notch-Dag chimeras (**B**) Luciferase reporter gene activity of Notch-Dag chimeras containing intact proteolytic switch and truncated switch with constitutive MMP sites (ΔCadΔSEA) upon addition of MMPs. Data shown are triplicate measurements from a representative experiment. Error bars represent the SEM of triplicate measurements and normalization is to no added MMP condition.

DOI: https://doi.org/10.7554/eLife.46983.008

(*Figure 2B*). These results suggest that the observed proteolytic switch-like behavior is relevant to regulation of dystroglycan's cleavage by MMPs in its native context, and that this assay can be used to further test hypotheses about regulation and potential modulation of proteolysis in dystroglycan.

## Shedding and modulation of shedding in diverse receptors detected by SNAPS

We next wanted to determine whether SNAPS could be used to detect membrane shedding of receptors that do not contain SEA-like domains. Proteolysis plays a major role in the function of cell surface receptors such as E-cadherin and receptor tyrosine kinases (RTKs) (*Katayama et al., 1994*; *Merilahti et al., 2017*; *Okamoto et al., 1999*), and dysregulation of proteolysis in these receptors is, for instance, linked to cancer pathogenesis (*Arribas et al., 2011*; *Brouxhon et al., 2014*; *Katayama et al., 1994*; *Okamoto et al., 1999*) and resistance to kinase inhibitor treatment (*Colomer et al., 2000*; *Leitzel et al., 1995*; *Miller et al., 2016*). Moreover, therapeutic antibodies such as Herceptin (*Molina et al., 2001*) have been developed that block proteolysis of the RTK HER2 and similar proteolysis blocking antibodies have been developed for the MICA immune receptor to block cancer cells from evading the immune system (*Ferrari de Andrade et al., 2018*). An assay with well-controlled input and outputs that could detect proteolysis and therapeutic efforts to modulate proteolysis would provide a valuable tool.

Unlike the aforementioned receptors with putative protease sites housed in structured SEA/SEA-like domains, the protease sites responsible for receptor shedding in cadherins and RTKs map to a putatively unstructured region between a structured repeat and the transmembrane region

(*Cho et al., 2003*; *D'Huyvetter et al., 2017*; *Franklin et al., 2004*; *Harrison et al., 2011*). These receptors might be expected to have higher basal levels of proteolysis and proteolytic regulation mechanisms distinct from SEA-like domain containing receptors; however, an assay that can detect proteolysis in such receptors could provide a starting point to test hypotheses about other potential mechanisms to regulate shedding, such as disruption of dimerization interfaces. We made chimeras of 10+ diverse cell surface receptors including HER and TAM family RTKs, E-cadherin, MICA, and CD44 (*Figure 3*, *Figure 4* and *Figure 1—figure supplement 5*) but focused our attention on the receptor tyrosine kinase HER2 and the cell adhesion receptor E-cadherin due to the well-established roles of proteolysis in their functions and the availability of extracellular antibodies that modulate receptor function with established (Herceptin) and potential (DECMA-1) therapeutic utility.

We used SNAPS to determine if we could measure: 1) basal proteolysis in a Notch-HER2 chimera and 2) its predicted modulation by Herceptin. Proteolysis of HER2 leads to a soluble ectodomain which is a prognostic biomarker as well as a membrane-tethered kinase domain with dysregulated activity (*Arribas et al., 2011*). We constructed a Notch-HER2 chimera replacing the Notch NRR with the HER2 ectodomain (*Figure 3A*). As expected, we observe significant basal proteolysis when the chimera is co-cultured with MS5 cells alone, in contrast to Notch (*Figure 3B*). Interestingly, like Notch, we observe a reproducible enhancement of signaling when Notch ligands are present that is observed in most RTKs tested (*Figure 1—figure supplement 5*). The relevance of this enhanced proteolysis when exposed to forces of cell to cell contact will be interesting to probe in future studies, but suggests that exposure of cryptic protease sites could contribute to proteolytic regulation in these receptors, perhaps by altering conformations of dimers. The monoclonal antibody trastuzumab

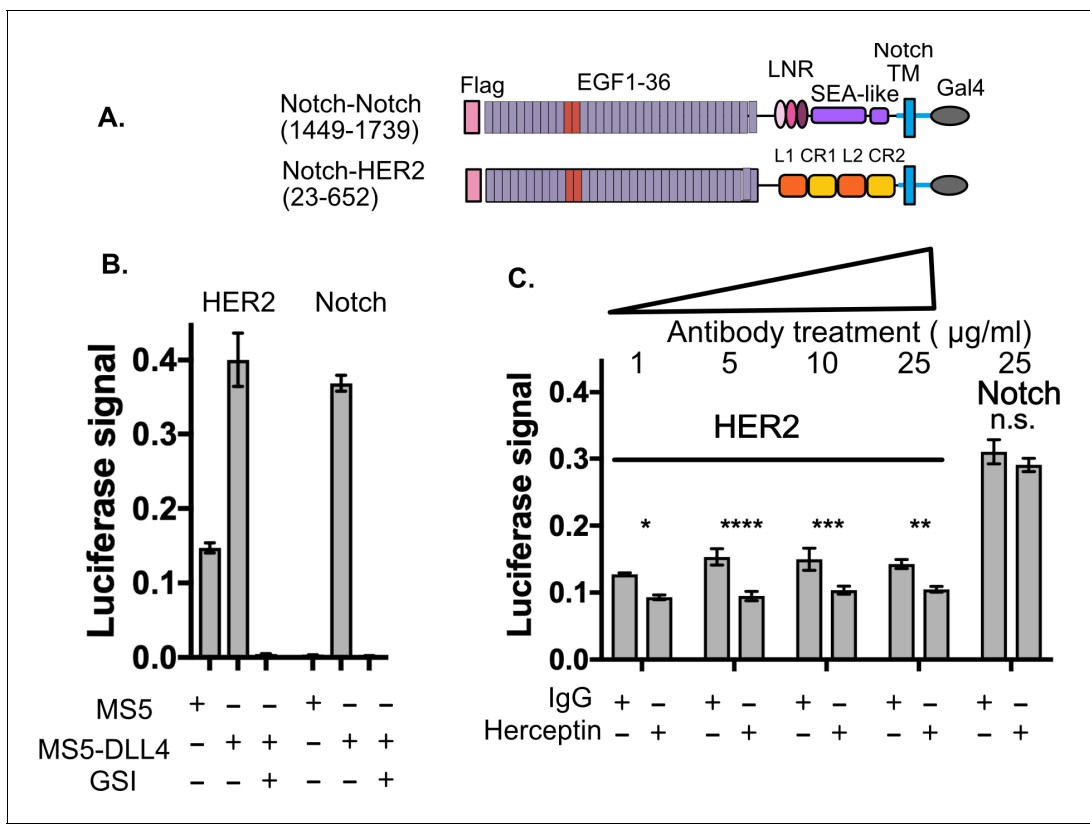

**Figure 3.** SNAPS detects HER2 shedding and shedding modulation by Herceptin. (**A**) Chimera constructs for Notch and HER2 (human epidermal growth factor receptor 2. Protein domains are color coded and labeled. (**B,C**) SNAPS assay measuring effect of Herceptin on basal signaling of HER2-Notch chimeras. Data shown are triplicate measurements from a representative experiment, error bars are SEM. (**B**) Untreated cells in co-culture with MS5 or MS5-DLL4 cells ± GSI for reference. (**C**) HER2-Notch chimera expressing cells co-cultured with MS5 cells were treated with 1–25 ug/ml Herceptin or IgG control. Statistical significance was determined with a two-way ANOVA followed by a post-hoc Bonferroni test. ****: p<0.0001, ***: p<0.001, **: p<0.01 *: p<0.02.
DOI: https://doi.org/10.7554/eLife.46983.009

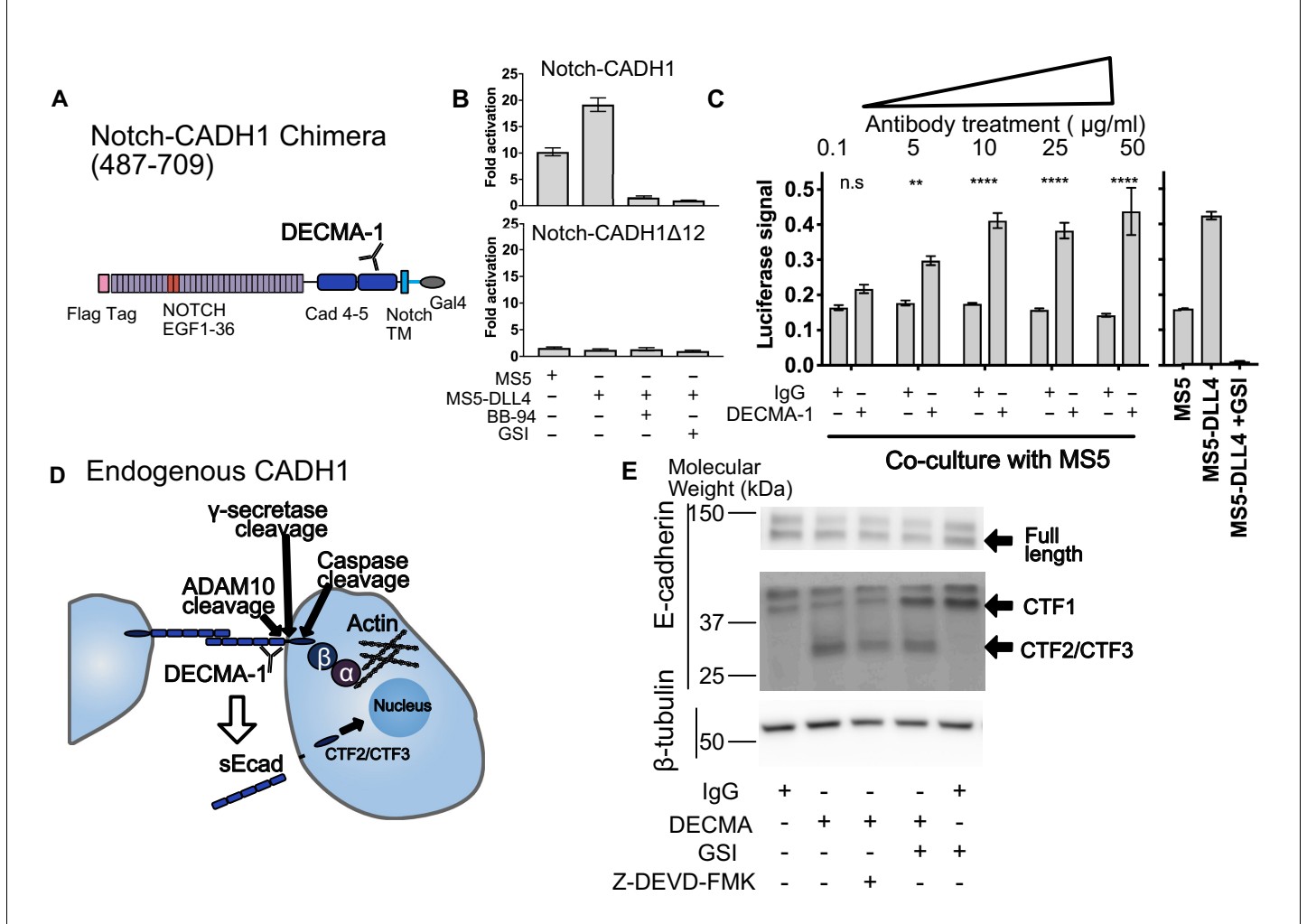

**Figure 4.** SNAPS reveals that E-cadherin proteolysis is a likely mechanism for DECMA-1 disruption of cell-cell adhesion. (**A**) Scheme of Notch-E-cadherin (CADH1) chimera in which cadherin repeats 4 and 5 replace the Notch NRR. (**B**) SNAPS assay on Notch-CADH1 chimera (top) and a construct with 10 amino acids containing putative cleavage sites removed (bottom). Assay normalized for GSI treatment. (**C**) SNAPS assay in which Notch-CADH1 chimeras were co-cultured with MS5 cells and treated with increasing concentrations of IgG control or DECMA-1. Raw luciferase signal shown. Untreated Notch CADH1 chimera shown on right for reference. Data shown are triplicate measurements from a representative experiment, error bars are SEM. Statistical significance was determined with a two-way ANOVA followed by a post-hoc Bonferroni test. ****: p<0.0001, ***: p<0.001, **:p<0.01 *: p<0.02. (**D**) Schematic of E-cadherin in its native context, with previously observed ADAM10, intramembrane, and caspase cleavage locations denoted as well as epitope of DECMA-1 binding. Beta-catenin (β) binds to cadherin's intracellular tail and can be translocated to the nucleus when E-cadherin is proteolyzed. (**E**) Western blot of E-cadherin cleavage products in MCF7 cells upon treatment with IgG control or DECMA-1 and inhibitors of gamma-secretase (GSI) and caspase cleavages (Z-DEVD-FMK).
DOI: https://doi.org/10.7554/eLife.46983.010

The following figure supplement is available for figure 4:

**Figure supplement 1.** DECMA-1 additional quantification.
DOI: https://doi.org/10.7554/eLife.46983.011

(Herceptin), used to treat HER2+ breast cancer (*Baselga et al., 1996*; *Pegram et al., 1998*) has been shown to block proteolysis of the HER2 receptor tyrosine kinase as part of its mechanism of action (*Molina et al., 2001*). Therefore, we tested whether Herceptin could modulate the basal proteolysis of HER2 observed in the Notch-HER2 chimeras. We treated HER2-chimera expressing cells with increasing concentrations of Herceptin or an IgG control (*Figure 3C*). We observed reproducible and statistically significant decreases in proteolysis in cells treated with Herceptin as compared to the IgG control (*Figure 3C*). Proteolysis was reduced up to 40%. Although the effects are the

same at all concentrations, there is no inhibition of the Notch construct, demonstrating specificity. While not surprising, the fact that Herceptin modulation of proteolysis in Notch-HER2 chimeras is detected serves as further validation of the assay.

We next used SNAPS to measure proteolysis of the cell-adhesion receptor E-cadherin. E-cadherin forms homotypic dimers between cells to facilitate cell adhesion. Proteolysis not only breaks the intercellular adhesion but also generates a soluble ectodomain that can activate RTK signaling pathways and alters localization of beta-catenin, which binds to the E-cadherin intracellular tail, from the membrane to the nucleus (*David and Rajasekaran, 2012*; *Maretzky et al., 2005*). We constructed Notch-cadherin chimeras comprised of the two cadherin repeats closest to the transmembrane either including or lacking the sequence containing putative cleavage sites between the terminal cadherin repeat and the membrane (*Figure 4A*). The observed 10-fold increase in basal signaling (co-culture with MS5 cells) over Notch was abrogated by treatment with protease inhibitors and when the linker containing putative ADAM10 sites was removed, suggesting that the signal is due to shedding. (*Figure 4B*).

Like HER2, antibodies that recognize E-cadherin's ectodomain have been developed to block its function. DECMA-1 is a function-blocking E-cadherin antibody known to break cell-cell contacts and reduce tumorigenesis in mice (*Brouxhon et al., 2013*). However, its mechanism of breaking cell-adhesions has remained elusive; the antibody binds to E-cadherin at the interface of the last two cadherin repeats (EC4 and EC5) near the membrane, but the N-terminal repeats EC1 and EC2 are responsible for the homotypic interactions presumed to be disrupted by the antibody. We hypothesized that DECMA-1 might affect E-cadherin shedding since the antibody epitope maps to the 'proteolysis region' of E-cadherin. Thus, we next tested the effects of DECMA-1 on proteolysis of the Cadherin-Notch chimera.

In the absence of antibody or when treated with IgG control antibodies, the Notch-cadherin chimera, in which cadherin repeats EC4 and EC5 have replaced the Notch proteolytic switch, displays a moderate level of constitutive proteolysis and a two-fold increase in activity in response to DLL4 expressing cells (*Figure 4B*). When the cells are treated with DECMA-1, we observe a dose-dependent increase in the basal level of signaling, in comparison to control antibody, almost to the level of ligand induced signaling (*Figure 4C*). The apparent EC50 of DECMA-1 measured by the assay is ~0.8 µg/mL (*Figure 4—figure supplement 1*). These data suggest that the mechanism of DECMA-1 breaking adhesive contacts could, in part, be due to increased shedding of the receptor from the membrane.

To provide further validation of the SNAPS assay and further explore the insight that proteolysis may provide the mechanism of DECMA-1's disruption of cell adhesions instead of the expected disruption of homotypic contacts, we tested DECMA-1's effects on proteolysis of endogenous E-cadherin in MCF7 cells. MCF7 cells were treated with DECMA-1 or IgG control in the presence or absence of proteolysis inhibitors. E-cadherin is known to be cleaved by at least three enzymes (*Figure 4D*). The first is ectodomain shedding by ADAM10 to form CTF1, which then makes the molecule a substrate for intramembrane cleavage by gamma-secretase to form CTF2. Finally, the intracellular domain of E-cadherin is known to be cleaved by Caspase-3 to generate CTF3, and this cleavage is enhanced in apoptotic cells. We observe a species at 37 kDa in all conditions (*Figure 4E*), which has been previously assigned as CTF1 (*Ferber et al., 2008*). When cells are treated with gamma-secretase inhibitors, the intensity of CTF1 increases, further confirming the assignment of this band as CTF1 (*Ferber et al., 2008*). The constitutive ectodomain shedding observed is consistent with the basal levels of proteolysis observed in the SNAPS assay. Strikingly, a lower molecular weight species of ~28 kDa appears when cells are treated with DECMA-1, consistent with DECMA-1 enhancing proteolysis of E-cadherin. This could be CTF2 or CTF3 according to previous studies (*Ferber et al., 2008*; *Steinhusen et al., 2001*). When cells are treated with DECMA-1 and a caspase three inhibitor, the band decreases about ~50% (quantification from two western blots shown in *Figure 4—figure supplement 1*), underscoring that it could be the caspase cleaved product. The intensity of the band also decreases 40% (*Figure 4—figure supplement 1*) when cells are treated with gamma secretase inhibitors, along with a concomitant increase in CTF1. Regardless of whether the fragment in CTF2 or CTF3, the result of DECMA-1 increasing proteolysis of endogenous E-cadherin in a relevant cancer cell line not only provides validation of using the SNAPS assay to study proteolysis in diverse receptors but also provides new mechanistic insight into how DECMA-1 functionally disrupts cell to cell adhesion.

## Discussion

Notch's proteolytic switch has been exploited to develop conformation-specific modulatory antibodies and harnessed for synthetic biology applications to turn on transcription in response to any desired cell to cell contact (*Agnusdei et al., 2014*; *Aste-Amézaga et al., 2010*; *Falk et al., 2012*; *Gordon et al., 2015*; *Li et al., 2008*; *Qiu et al., 2013*; *Roybal et al., 2016*; *Tiyanont et al., 2013*; *Wu et al., 2010*). Thus, we created SNAPS utilizing well-understood stimuli and responses of Notch signaling to identify novel proteolytic switches and probe shedding of a wide range of cell-surface receptors. Using this assay, we find that Notch's mechanism of proteolytic regulation via conformational control of a cryptic protease site is not a unique phenomenon and is rather a potentially common mechanism of control for several SEA-like domain-containing receptors that share structural homology to Notch. Moreover, shedding of transmembrane proteins such as HER2, AXL, CD44, and PCDH12 was detected with the assay, enabling new hypothesis generation about proteolytic regulation and modulation. SNAPS can also be used to screen for modulators of proteolysis; we observe that Herceptin treatment causes significant decreases in basal proteolysis of HER2, while DECMA-1 treatment results in substantial increases in basal proteolysis of E-cadherin. These results reveal new mechanistic insights into DECMA-1's function in breaking cellular adhesions.

### New proteolytic switches for synthetic biology

Our studies revealed that most receptors containing juxtamembrane SEA-like domains are robustly shed from the cell-surface and that several of them behave as proteolytic switches, only becoming sensitive to proteolysis when 'induced' with forces derived from cell-cell contact. We were struck by the fact that almost all of the receptors harboring SEA/SEA like domains in tandem with neighboring domains showed a similar switch-like behavior in the intercellular signaling assay, despite having neighboring domains with very different predicted structural characteristics. In Notch, the neighboring LNR domain is disulfide-rich and binds calcium, with little to no secondary structure (*Figure 1E*). Dystroglycan and the Protocadherins have neighboring cadherin-like domains, characterized by high β-strand content, while EpCAM and Trop2 have a cysteine-rich thyroglobulin domain. The existing crystal structures of several of these domains also reveal differential modes of interaction with the SEA/SEA-like domain. For example, in the EpCAM and NRR structures, the neighboring domain contacts the α-helix in close proximity to the β-strand containing putative proteolytic sites. In contrast, the cadherin-like domain interacts with the opposite face of the SEA-like domain in the Protocadherin-15 structure (*Figure 1E*). These different modes of interdomain interactions suggest that the proteolytic switches may have different propensities to 'switch on' as well as potentially different requirements for direction of applied force. Future studies probing comparative anatomy of putative proteolytic switches may reveal whether the structural differences are a consequence of cellular context; for example receptor involved in intercellular versus ECM interactions. On the other hand, the SEA-like domains lacking structured neighboring domains exhibit constitutive signaling, likely due to a more dynamic domain where protease site exposure occurs more frequently.

Synthetic biology applications that aim to induce transcription of a desired gene in response to cell to cell contact might benefit from proteolytic switches with different characteristics from the NRR of Notch. For example, in CAR-T applications, perhaps a switch that requires more 'force' to switch on could be used to distinguish an epitope that is presented on a tumor with a stiff ECM compared to a normal cell. Moreover, the smaller and structurally simpler design of the Cadherin-like neighboring domains of dystroglycan and protocadherin-15 might permit more facile trafficking and expression for certain applications. Finally, constitutively proteolyzed MUC1 and IA2 exhibit much higher expression/rates of proteolysis and may provide opportunities for engineering more robust switches when paired with a neighboring domain.

### Targeting proteolytic switches and shedding with therapeutics

Notch's proteolytic switch has been specifically targeted with both inhibitory and activating antibodies, suggesting that similar strategies could be successful for other receptors harboring proteolytic switches that are dysregulated in disease. While the proteolytic switches identified here need to be validated to determine if exposure of cryptic protease sites is physiologically relevant, validation experiments here and in a recent preprint (*Hayward and Gordon, 2018*) validates that dystroglycan protease sites are be conformationally controlled in the native receptor. Moreover, high levels of

MMPs and thus dystroglycan cleavage have been observed in muscle biopsies of muscular dystrophy patients (*Matsumura et al., 2005*), and treatment of muscular dystrophy mouse models with broad spectrum metalloprotease inhibitors has been shown to ameliorate symptoms in a muscular dystrophy mouse model (*Kumar et al., 2010*). The dystroglycan proteolytic switch might offer a receptor-specific therapeutic target in diseases where MMP cleavage is dysregulated. Moreover, SNAPS was also able to detect shedding and modulation of shedding in receptors that do not contain SEA-like domains, suggesting that the assay can provide a platform to screen modulators of shedding of diverse receptors.

## Exposure of cryptic protease site may be a common mechanotransduction mechanism

In this study, mechanical force derived from intercellular contact is applied to cell-surface receptors to identify cryptic protease sites. Many but not all receptor chimeras exhibited increased signaling in the presence of forces derived from intercellular contact. While mechanical force may not play a role in the function of some receptors studied here, several of the receptors probed have been implicated in mechanosensing. Like Notch (*Gordon et al., 2015*; *Langridge and Struhl, 2017*; *Parks et al., 2000*), E-cadherin (*Schwartz and DeSimone, 2008*) and Protocadherin-15 are involved in intercellular adhesions and transmission of mechanical stimuli. Protocadherin-15, for example, is involved in sensing sound vibrations across stereocilia tip links in the process of hearing (*Kazmierczak et al., 2007*). Mechanical forces are also known to be sensed at adhesions of cells with the ECM, as ECM stiffness dictates multiple cellular processes such as cell migration (*Lo et al., 2000*) and stem cell differentiation (*Engler et al., 2006*). For example, the ECM receptor CD44 is hypothesized to sense the stiffness of the ECM resulting in increased cell migration (*Kim and Kumar, 2014*; *Razinia et al., 2017*). Additionally, the ECM receptor dystroglycan is thought to act as a shock absorber to protect the sarcolemma during muscle contraction (*Barresi and Campbell, 2006*). Finally, even receptors that do not reside at canonical force sensing structures of cells have been implicated in mechanosensing. The RTK AXL which binds to a secreted ligand Gas6 has been shown to be a rigidity sensor (*Yang et al., 2016*) and facilitate a decrease in cellular stiffness in lung cancer (*Iida et al., 2017*). Thus, our studies showing that many receptors exhibit increased proteolysis in response to mechanical forces suggest that proteolysis may be a common mechanism used by cells to communicate mechanical stimuli. This assay could be used in the future to measure how varying the mechanical microenvironment affects receptor proteolysis.

## Limitations/caveats of assay

In the chimeric signaling assay, putative regions of proteolysis are evaluated in the context of artificial linkages at their N- and C-termini as well as potentially non-native stimuli and non-physiological presentation of proteases. In most cases, a small region of a receptor was excised and inserted into a larger receptor, resulting in non-native links to Notch's ligand binding and transmembrane domains. One might expect the artificiality of the chimeras would result in a majority of chimeric receptors signaling either constitutively or not at all. However, several receptors exhibited 'switch-like' behavior, underscoring the validity of SNAPS and the modular nature of cell-surface receptors. The use of the Notch transmembrane domain in the chimeric receptors also introduces some caveats as the domain, together with membrane proteins such as tetraspanins (*Zimmerman et al., 2016*), likely associates with the Notch membrane-tethered proteases ADAM10 and ADAM17. Although many of the chimeras studied have been reported to be cleaved by ADAM10 and ADAM17, some receptors may not typically reside in close proximity to these proteases and therefore not normally be cleaved. However, these proteases are upregulated in many diseases suggesting that the cleavage observed in this assay may be biologically relevant in certain cellular contexts. Finally, the chimeric Notch signaling assay provides a stimulus for exposing protease sites involving a 2–5 pN force normal to the cell surface. While many of the receptors studied here are also involved in cell-cell contacts likely involving similar mechanical forces, many interact with the ECM or have soluble ligands and perhaps may not normally be exposed to mechanical allostery. The main goal, however, was to provide a means to determine the presence of cryptic protease sites regardless of mechanical sensitivity. Harnessing this assay to study proteolytic regulation mechanisms is more specific than using,

for instance, APMA to non-specifically activate metalloproteinases (*Ogata et al., 1995*; *Stetler-Stevenson et al., 1989*).

### Conclusions

We have identified several putative proteolytic switches using SNAPS. These findings may drive development of conformation-specific modulatory antibodies as well as find use in synthetic biology applications that use cell to cell contact to drive transcriptional events. Our results provide a starting point for determining whether mechanisms of proteolytic regulation observed here are relevant to the biology of a given receptor. The convenient stimulus and response to proteolysis can be used to make additional chimeras to move closer to the native system and discover more about proteolytic regulation in the native receptor. For example, the luciferase response can be measured when systematically replacing chimeric domains with native transmembrane domains, ligand recognition domains, and intracellular tails. We also demonstrate that this assay provides a convenient platform for evaluating modulators of proteolysis.

## Materials and methods

### Reagents

Recombinant DLL4, MMP-2, and MMP-9 were purchased from R and D Systems. Batimastat (BB-94) was purchased from Sigma Aldrich. Compound E (GSI) was purchased from Fisher Scientific (Catalog # AAJ65131EXD). DECMA-1 antibody was purchased from Sigma-Aldrich (Cat# U3254, RRID:AB_477600). Z-DEV-FMK was purchased from R and D Systems (FMK004). E-cadherin primary antibody was purchased from Thermo Fisher Scientific (Cat# 33–4000, RRID:AB_2533118). β-tubulin primary antibody was purchased from Sigma-Aldrich (Cat#T8328, RRID:AB_1844090). U2OS cells were purchased from ATCC (Cat# HTB-96, RRID:CVCL_0042). MS5 and MS5-DLL4 cells were a kind gift from Dr. Stephen Blacklow. 4-aminophenylmercuric acetate (APMA) was purchased from Sigma-Aldrich. Herceptin was purchased from MedChemExpress (HY-P9907).

### Cloning

An Nhe1 site was added in Notch between amino acids 1735 and 1736 near the transmembrane region in a previously described Notch1-Gal4 construct (*Andrawes et al., 2013*) in the pcDNA5 FRT/TO backbone containing an N-terminal Flag tag, an AvrII site between the last EGF-like repeat and NRR, and a Bsu36i restriction site C-terminal to Notch transmembrane domain. All of the constructs were cloned using In-Fusion (Clontech).

CD44 was cloned using CD44S pWZL-Blast (RRID:Addgene_19126). Dystroglycan was cloned from cDNA from Origene (Cat#: SC117393). mTFP was cloned from TS module from (RRID:Addgene_26021). AXL, MerTK, and Tyro3 were originally ordered as *E. coli* optimized gBlocks (IDT) for different constructs and then cloned into the Notch chimera using primers with In-Fusion ends. HER2 and HER4 DNA was a kind gift from Dr. Laurie Parker, from the ORF kinase library (Addgene). The remaining constructs were ordered as mammalian codon optimized gBlocks from IDT with In-Fusion ends.

### Cell culture

All cell lines were cultured in DMEM (Corning) supplemented with 10% FBS (Gibco) and 0.5% penicillin/streptomycin (Gibco). Cells were incubated at 37°C in 5% $CO_2$.

### Notch signaling assay

The Notch signaling assay was performed as described (*Gordon et al., 2015*). For co-culture assays, 0.1, 1, 2, or 10 ng chimera constructs were reverse transfected with reporter plasmids (50 ng Gal4 reporter plasmid and 1 ng PRL-TK reporter plasmid) in triplicate into U2OS cells in a 96-well plate. 24 hr post-transfection, MS5 cells or MS5 cells stably expressing DLL4 were plated on top of the U2OS cells with DMSO or drug (40 µM BB-94 or 1 µM GSI). 48 hr post-transfection, cells were lysed in passive lysis buffer (Promega). Lysate was added to a white 96 well half volume plate, and Dual-Luciferase Reporter Assay (Promega) was performed according to manufacturer's recommendation and read out on a Molecular Devices LMaxII[384] plate reader.

In assays using recombinant MMP-2 and MMP-9, activated MMP-2 or MMP-9 was diluted to 0.46 µg/mL in D10 media and added 36 hr post-transfection. Media was swapped out 38 hr post-transfection. Cells were lysed in passive lysis buffer 50 hr post-transfection and read out as previously described. For signaling assays using antibody, DECMA-1, Herceptin, or Sheep IgG control was added during the co-culture step 24 hr post-transfection.

Data shown in figures are triplicate measurements from a representative experiment, although the experiments were performed three times unless noted.

### Cell surface ELISA

100 ng of Notch chimera constructs were transfected into U2OS cells in a sterile opaque tissue culture-treated 96-well plate (Corning 353296) in triplicate. 24 hr post-transfection, cells were washed once with PBS and fixed using 4% PFA (Thermo Fisher 28906) for 20 min, then washed three times with PBS. Cells were blocked in TBS +5% milk for 1 hr. Then, Flag primary antibody (Sigma-Aldrich F1804) was added 1:250 in TBS +5% milk for 2 hr. Cells were washed three times for 5 min each with TBS +5% milk. The cells were then incubated 1:10,000 with an HRP secondary antibody for 1 hr before being washed five times for 5 min each with TBS. Chemiluminescent substrate was added for 1 min before reading out on a luminescence plate reader.

### Western blot

MCF-7 cells were plated in 24-well tissue culture plates. 24 hr later, they were serum starved with 0.1% FBS in DMEM with antibiotics. DECMA-1 (100 µg/mL), Sheep IgG control (100 µg/mL), GSI (1 uM), Z-DEV-FMK (50 µM) were also added at this time. DMSO was added as a control. Cells were lysed with RIPA containing protease inhibitors 24 hr after serum starvation and drug treatment. Cell lysates were run on a 4–20% SDS-PAGE gel with 2 mM sodium thioglycolate in the running buffer. The protein was then transferred to a nitrocellulose membrane using a Genie Blotter (Idea Scientific) and blocked with 5% milk in TBS-T. E-cadherin antibody was diluted 1:500, and the β-tubulin antibody was diluted 1:1000. A goat-anti mouse HRP conjugated antibody (Invitrogen) was used as a secondary antibody. Western blots were imaged using chemiluminescent buffer (Perkin Elmer Western Lightning Plus ECL) and the Amersham 600UV (GE) with staff support at the University of Minnesota-University Imaging Center.

### Data and materials availability

All data needed to evaluate the conclusions of this study are available in the paper or the Supplementary Materials.

## Acknowledgements

We thank Steve Blacklow, Kassidy Thompkins, Maria Ramirez, and Robert Evans III for helpful comments on the manuscript and Klaus Lovendahl for cloning the TAM receptor chimeras and some of the RTK constructs. We thank the Aihara lab for use of their fluorescence plate reader. We also thank Steve Blacklow and Jon Aster for the DLL4 stable cell lines and the Notch1-Gal4 construct. We would like to thank the Parker lab for the HER2 and HER4 cDNAs.

## Additional information

### Funding

| Funder | Grant reference number | Author |
| --- | --- | --- |
| National Institute of General Medical Sciences | R35 GM119483 | Wendy R Gordon |
| Pew Charitable Trusts | Pew Biomedical Scholar | Wendy R Gordon |
| National Cancer Institute | U54CA210190 | Eric J Aird<br>Wendy R Gordon |
| 3M | Graduate student fellowship | Eric J Aird |
| American Heart Association | Graduate student fellowship | Amanda N Hayward |

| NIGMS | T32GM008347 | Eric J Aird |

The funders had no role in study design, data collection and interpretation, or the decision to submit the work for publication.

## Author contributions
Amanda N Hayward, Formal analysis, Validation, Investigation, Visualization, Writing—original draft, Writing—review and editing; Eric J Aird, Formal analysis, Validation, Investigation, Visualization, Writing—review and editing; Wendy R Gordon, Conceptualization, Resources, Formal analysis, Supervision, Funding acquisition, Validation, Investigation, Visualization, Methodology, Writing—original draft, Project administration, Writing—review and editing

## Author ORCIDs
Amanda N Hayward https://orcid.org/0000-0002-9252-0526
Eric J Aird https://orcid.org/0000-0002-4873-042X
Wendy R Gordon https://orcid.org/0000-0001-7696-5560

## Decision letter and Author response
Decision letter https://doi.org/10.7554/eLife.46983.015
Author response https://doi.org/10.7554/eLife.46983.016

## Additional files

### Supplementary files
• Supplementary file 1. Amino acid sequences for all proteolysis domains used in Notch-X chimeras.
DOI: https://doi.org/10.7554/eLife.46983.012

• Transparent reporting form
DOI: https://doi.org/10.7554/eLife.46983.013

### Data availability
All data generated or analysed during this study are included in the manuscript and supporting files. Source data files have been provided for Figures 1 and 3 and 4.

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
