## [Decision Letter]

[Editors’ note: a previous version of this study was rejected after peer review, but the authors submitted for reconsideration. The first decision letter after peer review is shown below.]

Thank you for submitting your work entitled "A toolkit for studying cell surface shedding of diverse transmembrane receptors" for consideration by *eLife*. Your article has been reviewed by three peer reviewers, one of whom is a member of our Board of Reviewing Editors, and the evaluation has been overseen by a Reviewing Editor and a Senior Editor. The following individuals involved in review of your submission have agreed to reveal their identity: Rhett A Kovall (Reviewer #3).

Our decision has been reached after consultation between the reviewers. Based on these discussions and the individual reviews below, we regret to inform you that your work will not be considered further for publication in *eLife*.

All three reviewers found the paper interesting, and the SNAPS technique clever. They also could see how it could be a powerful tool to look for regulated shredding of cell surface proteins. The problem is that the authors have not yet taken this beyond an early proof of concept. As described in the individual reviews below, the reviewers could easily imagine further developments in this work being appropriate for *eLife* but there was a clear consensus that it is just too preliminary to have high priority as it stands.

*Reviewer #1:*

The paper by Hayward et al., describes the establishment of an assay called SNAPS (Synthetic Notch Assay for Proteolytic Switches), used to assess the juxtamembrane cleavage of cell surface proteins. It builds on the mechanism by which proteolytic cleavage of the Notch receptor is regulated by conformational changes induced by binding to its ligands. The core of SNAPS is to swap the region of Notch responsible for this regulated proteolysis for equivalent regions of other proteins and test their ability to mediate cleavage, regulated or not by ligand interactions.

This is a nice idea and the basic question of examining an underexplored topic – the regulation of proteolytic shedding – is important. This work does not, however, fully live up to its initial promise. The discovery of three other SEA-like domain containing proteins that show DLL-dependent proteolysis, whilst not very surprising because of the domain conservation, is interesting and suggests new ideas about how these are regulated. I like the discussion about how the different N-terminal domains influence regulation and could represent mechanistic diversity.

I find the subsequent results around non-SEA-like domain proteins a bit confusing and unsatisfying. The subtlety of the SNAPS construct and its requirement for ligand interaction is not really exploited, and in several cases the assay becomes simply one for detecting cleavage of a juxtamembrane domain of a candidate protein. In the case of dystroglycan, the WT extracellular domain is cleaved less well than a truncated version, suggesting that there might be some regulation or switching but this is not pursued. Finally, the proof of concept of modulating cleavage by Herceptin and DECMA-1 is again an interesting starting point but, in the case of Herceptin, not very surprising, and in the case of DECMA-1 intriguing but not pursued.

This paper would be clearer if the authors could distinguish the cases where the constructs are simply being used to assay cleavage, and the more interesting cases where they are providing evidence for regulated proteolytic switches. And also, if they could distinguish novel (and perhaps surprising) biological results, from those which are not biologically surprising but which, perhaps they can argue, act as significant proofs of concept for the overall approach.

*Reviewer #2:*

The manuscript by Gordon and co-workers describes a clever approach that harnesses mechanical force derived from intercellular contact to identify cryptic protease sites in cell-surface receptors. It builds on Wendy Gordons impressive structural studies of the so-called negative regulatory region (NRR) in Notch, a domain that undergoes conformational changes in response to force revealing an ADAM 10 cleavage site. In this work chimeric receptors are generated, replacing the NRR with putative proteolytic switch domains from other trans-membrane receptors. The read-out relies on the nuclear translocation of a GAL4 fused to the intracellular domain to activate a luciferase reporter. The cleaved molecules become substrates for γ-secretase, which releases the GAL4 to give a quantitative read-out.

Except in a few cases, proteolysis of receptors is an under-studied topic, so the assay is a valuable one. It identifies several receptors that contain cryptic protease cleavage sites, based on the criteria tested in the assay. All of the work appears very solid. They include various inhibitors to show the activity is dependent on the proteases and they go on to illustrate some uses, e.g. to screen for inhibitors. Altogether the work is original and conducted to a high standard.

The results demonstrate the usefulness of their assay. What is missing is either some context and evidence to support the biological importance of the regulation they have uncovered or further biophysical analysis to assess whether proteolysis of one/some of the receptors is indeed regulated mechanically.

Essential revisions:

Figure 2B: "wild-type dystroglycan with intact proteolytic switch shows low levels of the 31-kDa cleavage product". None of the smaller band is detected with the wild-type and none of the larger band in the mutated version. The authors should repeat the experiment adding the relevant inhibitors to confirm whether the fragments are produced by the indicated enzymes. Without these data it is hard to appreciate how the authors have arrived at their conclusions.

Figure 4A: Only a very modest effect is detected (less than 2-fold) and this is the same for all concentrations of the antibody used. How can they rule out a non-specific effect from the antibody-for example does it affect activity of any other chimeras?

As they mention, "some receptors may not typically reside in close proximity to these proteases and therefore not normally be cleaved." If they could provide evidence to demonstrate cleavage occurring in vivo (with endogenous proteins) it would significantly strengthen their story.

The authors comment that their findings "enabling new hypothesis generation about proteolytic regulation and modulation" of receptors. Can they elaborate on how they envisage proteolysis contribute to the regulation of the receptors; can they test some of these possibilities?

*Reviewer #3:*

The authors' manuscript describes a new method called SNAPS (Synthetic Notch Assay for Proteolytic Switches) to characterize the proteolysis of cell surface receptors, which are generally understudied. SNAPS takes advantage of the modularity and well-characterized molecular mechanism of the regulated intramembrane proteolysis (RIP) of the Notch receptor when it's in complex with its ligand. In this study, the authors make chimeric constructs that use the ligand binding domain of the Notch receptor fused to the extracellular domains of heterologous receptors, some of which contain SEA-like domains and adjacent regulatory domains, followed by the transmembrane region of the Notch receptor and an intracellular GAL4 domain that can activate the luciferase reporter when the chimera is cleaved in the presence of cocultured cells that express a Notch ligand (DLL4 in this case). The data suggests that SNAPS appears to be flexible enough to accommodate numerous different chimeric receptors and can be used to screen for reagents or small molecules that modulate proteolysis of cell surface receptors. Overall, I find this this manuscript to be interesting and novel, as well as concise and well-written, and addresses an understudied area of biology, providing a new tool that would be useful to researchers. If the authors were to address my relatively minor concerns summarized below I would be happy to recommend that their manuscript be published in *eLife*.

In the Discussion section, the authors describe the structural elements that are involved in regulating proteolysis from different SEA-like domains. With the limited structural data presented in Figure 1, it is very difficult to follow the authors' reasoning. I would suggest the inclusion of additional structure information to support their arguments in the Discussion section as either additional panels in Figure 1 or as a supplemental figure.

As the authors are aware, cellular Notch receptors can also be activated when recombinant ligands are coated on the surface of tissue culture plates. Does SNAP also work under these conditions? This information might be useful in the context of proteolysis for cell surface receptors that can't be co-cultured with cells expressing Notch ligands.

Have the authors tested whether adjacent regulatory domains can regulate the proteolysis of heterologous receptors, with or without SEA-like domains? This would be interesting to know in the context of synthetic biology, for example could you combine different adjacent regulatory domains with different receptors that are proteolyzed to design new chimeric receptors with new activities?

[Editors’ note: what now follows is the decision letter after the authors submitted for further consideration.]

Thank you for resubmitting your work entitled "A toolkit for studying cell surface shedding of diverse transmembrane receptors" for further consideration at *eLife*. Your revised article has been reviewed by three peer reviewers, one of whom is a member of our Board of Reviewing Editors, and the evaluation has been overseen by a Senior Editor.

You have addressed the comments of the initial reviews by adding some new data, referring to a related manuscript submitted elsewhere, and rewriting some of the text.

The newer version of the paper was felt to have the required significance to be appropriate for *eLife*. Nevertheless, the consensus view is that this paper is more appropriate as a 'tools and resources' article than a standard research paper. In summary, the SNAPs assay is potentially valuable resource that merits publication, but some additional modifications to the text are needed to reflect the refocusing of the manuscript on the assay itself. At the moment it conveys a mixed message.

The newly added text about dystroglycan is not so definitive. We suggest that this section should be modified to reflect the change of emphasis (eg commence the section by setting the context provided in the last sentence, namely: "investigate whether this assay can be used effectively to further test hypotheses about regulation and potential modulation of proteolysis by the SEA domains"). Explain that dystroglycan is being used as a model because the conformational regulation of proteolysis of the native dystroglycan protein in vitro has been explored in parallel studies

It would also be constructive to place the emphasis on the SEA/ proteolytic domain protecting against proteolysis rather than on its deletion enhancing proteolysis. This section is a bit confusing and lacking in clear relevance to the paper.

Re the herceptin treatment, Figure 3C. The authors do need to add a comment in subsection “Shedding and modulation of shedding in diverse receptors detected by SNAPS” about the fact that the effects are the same at all concentrations and highlight that there is no inhibition of the Notch construct.

Figure 4E should be quantified. The text indicates there is a decrease in intensity from CTF2/3 27kd band when the GSI or caspase inhibitors are added. The reproducibility file indicates that 3 replicates were performed so the data should be available.

In the reproducibility file, the authors refer to replicates performed on the same experiment as biological replicates, because they were replicate transfections. These might be better considered as technical replicates; the term biological replicates is best kept for experiments performed on separate days with separate preparations of transfection mixes etc. They do explain what has been done in the reproducibility document, but this is not evident from the figure legends or methods. For example, in Figure 1C, these data appear to be from one experiment as they are in "triplicate" but none of the experiments were repeated 3 times? Best practice would be to present data from 3 experiments performed on 3 different occasions. If this is not the case, the authors need to state explicitly what they are referring to as replicates: E.g. triplicates from a representative experiment?

They should also clarify the amount of DNA used e.g. in 1C. It is important that some information about the robustness is provided. What was the variability between experiments? The impression from Figure 1—figure supplement 2 is that more DNA is better up to 10ng. Or would they recommend that it is worth testing a range of concentrations? It would be helpful to make a statement about this and any other relevant technical details.

---

## [Author Response]

[Editors’ note: the author responses to the first round of peer review follow.]

We are grateful for the reviewers’ comments and suggestions, and our manuscript has been greatly improved by addressing the concerns. We were excited that the reviewers noted many strengths of the study: “All three reviewers found the paper interesting, and the SNAPS technique clever. They also could see how it could be a powerful tool to look for regulated shredding of cell surface proteins.” Multiple reviewers also pointed out that we were addressing an under-explored and important topic and we were glad they found the work “solid and conducted to a high standard”. The major concern of all three reviewers was that we did not provide sufficient validation of 1) the proteolytic switch behavior discovered for several receptors and 2) the detection of proteolysis and modulation of proteolysis in the non-proteolytic switch-like receptors. We provide the details in the point by point rebuttal below but briefly, we addressed this concern by: 1) providing/discussing a companion manuscript from our lab on the conformational regulation of proteolysis in dystrolycan which validates the conformational switch behavior observed for dystroglycan and 2) performing new experiments on endogenous E-cadherin in MCF7 cells in which observe that DECMA-1 increases proteolysis of E-cadherin, consistent with the results of the SNAPS assay. This not only validates a result from the SNAPS assay but provides new mechanistic insight into how DECMA-1 functions to disrupt intercellular adhesion.

Reviewer #1:The paper by Hayward et al., describes the establishment of an assay called SNAPS (Synthetic Notch Assay for Proteolytic Switches), used to assess the juxtamembrane cleavage of cell surface proteins. It builds on the mechanism by which proteolytic cleavage of the Notch receptor is regulated by conformational changes induced by binding to its ligands. The core of SNAPS is to swap the region of Notch responsible for this regulated proteolysis for equivalent regions of other proteins and test their ability to mediate cleavage, regulated or not by ligand interactions.This is a nice idea and the basic question of examining an underexplored topic – the regulation of proteolytic shedding – is important. This work does not, however, fully live up to its initial promise. The discovery of three other SEA-like domain containing proteins that show DLL-dependent proteolysis, whilst not very surprising because of the domain conservation, is interesting and suggests new ideas about how these are regulated. I like the discussion about how the different N-terminal domains influence regulation and could represent mechanistic diversity.

We are glad that you find the work important and the discovery of the new proteolytic switches interesting.

I find the subsequent results around non-SEA-like domain proteins a bit confusing and unsatisfying. The subtlety of the SNAPS construct and its requirement for ligand interaction is not really exploited, and in several cases the assay becomes simply one for detecting cleavage of a juxtamembrane domain of a candidate protein.

We completely agree that the non-SEA part of the paper became a laundry list of results. In part, it was meant to provide results a “screen” so that others could see if proteolysis could be detected in their favorite protein. To alleviate this concern, we focused the results on two non-SEA receptors- HER2 and DECMA-1- and moved the rest to Figure 1—figure supplement 5.

In the case of dystroglycan, the WT extracellular domain is cleaved less well than a truncated version, suggesting that there might be some regulation or switching but this is not pursued.

In parallel, our lab has been pursuing in-depth studies of the conformational regulation of dystroglycan. We have now provided a copy of this manuscript, under revision at Molecular and Cellular Biology, and also added discussion of these results (and citation of the paper) in the revised manuscript. In this paper, we found that dystroglycan containing an intact “proteolytic switch” domain is resistant to MMP proteolysis. Domain truncations and destabilizing muscular dystrophy and cancer related mutations lead to enhanced proteolysis. Moreover, we found correlations between proteolysis and cell adhesion/migration phenotypes, underscoring the physiologic relevance of proteolysis in dystroglycan. We feel that these results provide a clear validation of our SNAPS results.

Finally, the proof of concept of modulating cleavage by Herceptin and DECMA-1 is again an interesting starting point but, in the case of Herceptin, not very surprising, and in the case of DECMA-1 intriguing but not pursued.

We completely agree and performed new experiments to address this point. Regarding Herceptin, we have added language in the text to state that the fact that we observe a decrease in proteolysis is not surprising and, in fact, provides some level of validation of the assay. Thank you for making this point. Regarding DECMA-1, we decided to pursue this interesting result and determine if DECMA-1 also increased proteolysis in endogenous E-cadherin. We treated MCF7 cells with DECMA-1 or IgG control and blotted for endogenous E-cadherin C-terminal domain to detect proteolytic products. First of all, we observe constitutive E-cadherin shedding by ADAM10, which is consistent with the reasonably high basal levels of shedding observed in our chimeras. Second, we observe the drastic appearance of smaller proteolytic fragments when cells were treated with DECMA-1, corresponding either to the intramembrane cleaved or caspase cleaved E-cadherin that have been previously observed in the literature. These data are presented in Figure 4E. We believe that this both further validates the SNAPS assay and provides exciting new insight into DECMA-1 blocking antibody function.

This paper would be clearer if the authors could distinguish the cases where the constructs are simply being used to assay cleavage, and the more interesting cases where they are providing evidence for regulated proteolytic switches. And also, if they could distinguish novel (and perhaps surprising) biological results, from those which are not biologically surprising but which, perhaps they can argue, act as significant proofs of concept for the overall approach.

We agree and attempted to address this by moving the constructs simply being used to assay cleavage to Figure 1—figure supplement 5 and new language in the text.

Reviewer #2:

The manuscript by Gordon and co-workers describes a clever approach that harnesses mechanical force derived from intercellular contact to identify cryptic protease sites in cell-surface receptors. It builds on Wendy Gordons impressive structural studies of the so-called negative regulatory region (NRR) in Notch, a domain that undergoes conformational changes in response to force revealing an ADAM 10 cleavage site. In this work chimeric receptors are generated, replacing the NRR with putative proteolytic switch domains from other trans-membrane receptors. The read-out relies on the nuclear translocation of a GAL4 fused to the intracellular domain to activate a luciferase reporter. The cleaved molecules become substrates for γ-secretase, which releases the GAL4 to give a quantitative read-out.Except in a few cases, proteolysis of receptors is an under-studied topic, so the assay is a valuable one. It identifies several receptors that contain cryptic protease cleavage sites, based on the criteria tested in the assay. All of the work appears very solid. They include various inhibitors to show the activity is dependent on the proteases and they go on to illustrate some uses, e.g. to screen for inhibitors. Altogether the work is original and conducted to a high standard.

Thank you for your assessment that the work is clever, solid, and conducted to a high standard.

The results demonstrate the usefulness of their assay. What is missing is either some context and evidence to support the biological importance of the regulation they have uncovered or further biophysical analysis to assess whether proteolysis of one/some of the receptors is indeed regulated mechanically.

We have addressed the biological importance as discussed in points 1-3 and 1-4 above- by discussing (and providing the manuscript to you) a parallel study on dystroglycan proteolysis and performing new experiments on endogenous E-cadherin. We have not been able to assess whether the proteolysis is regulated mechanically. We have provided a Supplemental figure showing that the assay also works with plated Notch ligand, which is another mode of activating Notch by force, but the single molecule experiments for some molecules are ongoing in our lab and the hope is that this work will inspire others to pursue such studies.

Essential revisions:Figure 2B: "wild-type dystroglycan with intact proteolytic switch shows low levels of the 31-kDa cleavage product". None of the smaller band is detected with the wild-type and none of the larger band in the mutated version. The authors should repeat the experiment adding the relevant inhibitors to confirm whether the fragments are produced by the indicated enzymes. Without these data it is hard to appreciate how the authors have arrived at their conclusions.

We removed this data from the SNAPS paper to alleviate confusion. These constructs were taken from the companion study provided to you and were a small subset of truncations/mutations used to interrogate conformational regulation of dystroglycan proteolysis. We believe the data is better represented in the other paper where there is more context and rationale for them. We did, however, leave the SNAPS assay experiments treating the full-length and truncated Notch chimeras with exogenous MMPs because it nicely recapitulates the fact that the truncated version is more sensitive to proteases than the full-length, which was what was observed in the other paper.

Figure 4A: Only a very modest effect is detected (less than 2-fold) and this is the same for all concentrations of the antibody used. How can they rule out a non-specific effect from the antibody-for example does it affect activity of any other chimeras?

We cannot explain why we did not see a nice dose response like in the case of DECMA-1 (source of Herceptin, need to serum starve, blocking proteolysis only a partial effect even in endogeous protein?). However, the effect was statistically significant in every experiment (performed at least four times). In the revised manuscript, we now provide data from the same experiment showing the ligand-induced Notch signaling is not affected by Herceptin treatment.

*As they mention, "some receptors may not typically reside in close proximity to these proteases and therefore not normally be cleaved." If they could provide evidence to demonstrate cleavage occurring* in vivo *(with endogenous proteins) it would significantly strengthen their story.*

Since the first version of the paper showed all of the non-SEA receptors in Figure 2, we did not adequately discuss the existing strong literature evidence for proteolysis of endogenous receptors. We now do provide more discussion of HER2 and E-cadherin proteolysis. Moreover, the new experiment we performed on endogeous E-cadherin in MCF7 cells (Figure 4E) shows appreciable constitutive shedding of E-cadherin.

The authors comment that their findings "enabling new hypothesis generation about proteolytic regulation and modulation" of receptors. Can they elaborate on how they envisage proteolysis contribute to the regulation of the receptors; can they test some of these possibilities?

In the literature, proteolysis has been established to control cell communication by: (1) Breaking adhesions (2) Releasing products that are involved in signaling inside of the cell OR releasing soluble ectodomains that agonize or antagonize other signaling pathways (3) removing epitopes required for recognition by the immune system or other factors (4) regulating cell-surface availability. While the Notch chimeras do not show how proteolysis is aiding in cell communication for native receptors, it provides a way to measure proteolysis, with the potential to add back components of the native system to determine relevant mechanisms. In our parallel manuscript, we show that proteolysis of dystroglycan correlates with cell migration thus proteolysis acts to break its adhesion with the extracellular matrix molecule laminin. In this study, we also link proteolysis to breaking of cell adhesions of E-cadherin; DECMA-1 increases proteolysis and also is known to break cell adhesions

Reviewer #3:The authors' manuscript describes a new method called SNAPS (Synthetic Notch Assay for Proteolytic Switches) to characterize the proteolysis of cell surface receptors, which are generally understudied. SNAPS takes advantage of the modularity and well-characterized molecular mechanism of the regulated intramembrane proteolysis (RIP) of the Notch receptor when it's in complex with its ligand. In this study, the authors make chimeric constructs that use the ligand binding domain of the Notch receptor fused to the extracellular domains of heterologous receptors, some of which contain SEA-like domains and adjacent regulatory domains, followed by the transmembrane region of the Notch receptor and an intracellular GAL4 domain that can activate the luciferase reporter when the chimera is cleaved in the presence of cocultured cells that express a Notch ligand (DLL4 in this case). The data suggests that SNAPS appears to be flexible enough to accommodate numerous different chimeric receptors and can be used to screen for reagents or small molecules that modulate proteolysis of cell surface receptors. Overall, I find this this manuscript to be interesting and novel, as well as concise and well-written, and addresses an understudied area of biology, providing a new tool that would be useful to researchers. I would be happy to recommend that their manuscript be published in eLife.

Thank you for these comments!

In the Discussion section, the authors describe the structural elements that are involved in regulating proteolysis from different SEA-like domains. With the limited structural data presented in Figure 1, it is very difficult to follow the authors' reasoning. I would suggest the inclusion of additional structure information to support their arguments in the Discussion section as either additional panels in Figure 1 or as a supplemental figure.

We are actually writing a review on this topic! We have added more information about adjacent domains and interdomain buried surface to the Figure legend.

As the authors are aware, cellular Notch receptors can also be activated when recombinant ligands are coated on the surface of tissue culture plates. Does SNAP also work under these conditions? This information might be useful in the context of proteolysis for cell surface receptors that can't be co-cultured with cells expressing Notch ligands.

The assay does work with plated ligand. We have added Figure 1—figure supplement 4 to address this.

Have the authors tested whether adjacent regulatory domains can regulate the proteolysis of heterologous receptors, with or without SEA-like domains? This would be interesting to know in the context of synthetic biology, for example could you combine different adjacent regulatory domains with different receptors that are proteolyzed to design new chimeric receptors with new activities?

We have made some attempts at this and have not found the right combination of surface complementarity between adjacent domain and random SEA-like domain- but agree that this would be an attractive feature to synthetic biologists!

[Editors' note: the author responses to the re-review follow.]

You have addressed the comments of the initial reviews by adding some new data, referring to a related manuscript submitted elsewhere, and rewriting some of the text.

*The newer version of the paper was felt to have the required significance to be appropriate for* eLife*. Nevertheless, the consensus view is that this paper is more appropriate as a 'tools and resources' article than a standard research paper. In summary, the SNAPs assay is potentially valuable resource that merits publication, but some additional modifications to the text are needed to reflect the refocusing of the manuscript on the assay itself. At the moment it conveys a mixed message.*

Thank you for all of the helpful comments.

*The newly added text about dystroglycan is not so definitive. We suggest that this section should be modified to reflect the change of emphasis (eg commence the section by setting the context provided in the last sentence, namely: "investigate whether this assay can be used effectively to further test hypotheses about regulation and potential modulation of proteolysis by the SEA domains"). Explain that dystroglycan is being used as a model because the conformational regulation of proteolysis of the native dystroglycan protein* in vitro *has been explored in parallel studies.*

Thank you for the suggestions for making presentation of validation of SNAPS using dystroglycan as a model more clear. We have completely rewritten/trimmed this section.It would also be constructive to place the emphasis on the SEA/ proteolytic domain protecting against proteolysis rather than on its deletion enhancing proteolysis.This section is a bit confusing and lacking in clear relevance to the paper.

We have changed the emphasis of this discussion to protecting against proteolysis as suggested.

Re the herceptin treatment, Figure 3C. The authors do need to add a comment in subsection “Shedding and modulation of shedding in diverse receptors detected by SNAPS” about the fact that the effects are the same at all concentrations and highlight that there is no inhibition of the Notch construct.

Done.

Figure 4E should be quantified. The text indicates there is a decrease in intensity from CTF2/3 27kd band when the GSI or caspase inhibitors are added. The reproducibility file indicates that 3 replicates were performed so the data should be available.

For the inhibitor treatment of antibody treated cells, we actually performed two replicates, and have specified that in the reproducibility file. We performed 4 replicates of the experiment with DECMA-1 treatment compared to IgG control. We have added a graph to Figure 4—figure supplement1.

In the reproducibility file, the authors refer to replicates performed on the same experiment as biological replicates, because they were replicate transfections. These might be better considered as technical replicates; the term biological replicates is best kept for experiments performed on separate days with separate preparations of transfection mixes etc. They do explain what has been done in the reproducibility document, but this is not evident from the figure legends or methods. For example, in figure 1C, these data appear to be from one experiment as they are in "triplicate" but none of the experiments were repeated 3 times? Best practice would be to present data from 3 experiments performed on 3 different occasions. If this is not the case, the authors need to state explicitly what they are referring to as replicates: E.g. triplicates from a representative experiment?

We have added a statement to each Figure and to the Materials and methods section. Generally, the data in the figures represents triplicate measurements from a representative experiment. However, we did perform experiments at least three times and observed similar trends.

They should also clarify the amount of DNA used e.g. in 1C. It is important that some information about the robustness is provided. What was the variability between experiments? The impression from Figure 1—figure supplement 2 is that more DNA is better up to 10ng. Or would they recommend that it is worth testing a range of concentrations? It would be helpful to make a statement about this and any other relevant technical details.

We added a couple of sentences about our recommendation to test a range of concentrations, given that stimulation by ligand can be masked in receptors with unusually high cell surface expression.